# Multilevel geospatial analysis of factors associated with unskilled birth attendance in Ghana

**Vincent Bio Bediako**[1]*, **Ebenezer N. K. Boateng**[2], **Bernard Afriyie Owusu**[1], **Kwamena Sekyi Dickson**[1]

1 Department of Population and Health, University of Cape Coast, Cape Coast, Ghana, 2 Department of Geography and Regional Planning, University of Cape Coast, Cape Coast, Ghana

* vincent.bediako@stu.ucc.edu.gh

## Abstract

**Data Availability Statement:** The data used in this study are third party data from DHS (https://www.dhsprogram.com/data/available-datasets.cfm) and can be accessed following the protocol outlined in the Methods section.

### Background

Globally, about 810 women die every day due to pregnancy and its related complications. Although the death of women during pregnancy or childbirth has declined from 342 deaths to 211 deaths per 100,000 live births between 2000 and 2017, maternal mortality is still higher, particularly in sub-Saharan Africa and South Asia, where 86% of all deaths occur.

### Methods

A secondary analysis was carried out using the 2014 Ghana Demographic and Health Survey. A sample total of 4,290 women who had a live birth in the 5 years preceding the survey was included in the analysis. GIS software was used to explore the spatial distribution of unskilled birth attendance in Ghana. The Geographic Weighted Regression (GWR) was employed to model the spatial relationship of some predictor of unskilled birth attendance. Moreover, a multilevel binary logistic regression model was fitted to identify factors associated with unskilled birth attendance.

### Results

In this study, unskilled birth attendance had spatial variations across the country. The hotspot, cluster and outlier analysis identified the concerned districts in the north-eastern part of Ghana. The GWR analysis identified different predictors of unskilled birth attendance across districts of Ghana. In the multilevel analysis, mothers with no education, no health insurance coverage, and mothers from households with lower wealth status had higher odds of unskilled birth attendance. Being multi and grand multiparous, perception of distance from the health facility as not a big problem, urban residence, women residing in communities with medium and higher poverty level had lower odds of unskilled birth attendance.

**Funding:** The authors received no specific funding for their work.

**Competing interests:** The authors have declared that no competing interest exist.

**Abbreviations:** AICc, Akaike's Information Criterion; AOR, Adjusted Odds Ratio; EAs, Enumeration Areas; GDHS, Ethiopian Demographic and Health Survey; GWR, Geographically Weighted Regression; OLS (M), Ordinary Least Square (Model); SBA, Skilled Birth Attendant; WHO, World Health Organization.

## Conclusion

Unskilled birth attendance had spatial variations across the country. Areas with high levels of unskilled birth attendance had mothers who had no formal education, not health insured, mothers from poor households and communities, primiparous women, mothers from remote and border districts could get special attention in terms of allocation of resources including skilled human power, and improved access to health facilities.

## Background

Globally, about 810 women die every day due to pregnancy and its related complications [1]. Although the death of women during pregnancy or childbirth has declined from 342 deaths to 211 deaths per 100,000 livebirths between 2000 and 2017, maternal mortality is still higher, particularly in sub-Saharan Africa and South Asia, where 86% of all deaths occur. The lifetime risk of obstetric causes of maternal deaths in developing countries is 33 times higher than deaths in developed countries. Evidence depicts that 94 per cent of maternal and child deaths occur in low-and-middle-income countries, including Ghana [1]. However, Ghana has chalked major successes in its quest to reduce maternal and neonatal mortality. For instance, the maternal mortality ratio (MMR) has declined from 760 deaths per 100,000 live births in 1990 to 350/100,000 live births in 2010 and 319/100,000 livebirths in 2015 [2]. Similarly, neo-natal deaths (NMR) have steadily declined from 59 deaths per 1,000 live births in 1970 to 23 per 1,000 live births in 2019. These reductions are also due to the substantial increase in insti-tutional healthcare deliveries from 46% in 2003 to 78% in 2018 [2]. However, births assisted by skilled personnel in Ghana are below 70%, compared to the global 81 per cent coverage [3].

Skilled birth attendance (SBA) is defined as births assisted by healthcare trained profession-als such as doctors, midwives, and nurses to manage uncomplicated pregnancies, childbirth, the postpartum period (within 24 hours), and new-born health [4]. SBA has been identified as one of the surest ways of extenuating the risks of maternal deaths during pregnancy [3]. There-fore, it is recommended that SBA be relied upon throughout the continuum of care as it has clinical rationales–prevention of direct and indirect obstetric complications including haemor-rhage, eclampsia, obstetric fistula, obstructed labour, unsafe abortions [5, 6], and HIV and malaria, respectively. Further, neonatal deaths are lowest in countries with the highest cover-age of skilled birth attendance [7], and it is found that SBA reduces stillbirth by 23% [8].

In Ghana, appreciable progress has been made to promote SBA. For instance, the National Health Insurance Scheme (NHIS) covers SBA, and pregnant women are further exempt from paying NHIS premiums [9]. Considerable improvements have also been achieved in Ghana's Universal access to healthcare as Community-based Health Planning Services (CHPS) have been extended to reach hitherto remote areas in the country. Therefore, it is not surprising that SBA has increased from 44% in 1993 to 68% in 2011 [10]. However, MMR of 319/100,000 and NMR of 30 deaths per 1,000 live births are still unexpectedly high, and a major contributor of MMR and NMR is unskilled birth attendance, defined as births assisted by a traditional birth attendant (TBA), traditional health volunteer, community/village health volunteer, neighbours/ friends/relatives, and other untrained individuals [11]. Such women are usually assisted by either traditional birth attendants or relatives and friends [12], and this has dire consequences for both the mother and her unborn child or children. Unskilled birth atten-dance is undoubtedly a major drawback to achieving SDG 3 target 3.1, aiming to reduce the global maternal mortality ratio to less than 70 per 100,000 [3].

Several factors have been identified in Ghana to influence women's decision to utilize unskilled birth attendance. These factors include low socioeconomic status, no national health insurance (NHIS) coverage, residing in rural areas, higher-order parity, ethnicity, traditional religion, previous contact with the health system, and low levels of education [13–15]. Budu [13] found that as parity increased, home delivery was more likely to occur, and women with no NHIS were more likely to deliver at home. In other reports, Apanga and Awoonor-Williams [16] found that physical inaccessibility to health care, socio-cultural and economic factors, and health care system challenges were the major causes of low levels of skilled birth attendance in Ghana. In his equity analysis also, Zere et al. [17] showed that many maternal health services, including access to SBA and caesarean section services, healthcare facilities, use of modern contraceptives, and intermittent treatment for malaria, are not equitably distributed as women in wealthier households are more likely to utilize Maternal and Child Health (MCH) services, compared to their counterparts from poorer households. Evidence of North-South variations in SBA has been reported as well.

Skilled birth attendance (SBA) uptake is higher in southern Ghana [13, 18, 19] compared to Northern parts of Ghana [20]. Such variations and uneven use of services blurs national progress and continuously perpetuates inequalities in health. Cues must be taken because Ghana failed to achieve some Millennium Development Goals (MDGs) targets, including those related to maternal health. We, therefore, analyse the current Demographic and Health Survey data of Ghana to determine individual, community, and regional levels factors predicting unskilled birth attendance in Ghana. We further contribute to the maternal and child health literature in Ghana by highlighting hotspots for unskilled birth attendance and the factors associated with such observations. Previous geospatial studies have focused on inadequate variables [21] or focused on skilled birth attendance [22]. Understanding the predictors of unskilled birth attendance and its geospatial coverage in Ghana is of high relevance to both local and public health understanding of unskilled birth attendance and highlight areas where national interventions will yield cost-effective results.

## Methods

### Source of data

The Ghana Demographic and Health Survey (GDHS) used a standard Demographic and Health Survey (DHS) model questionnaire developed by the Measure DHS programme. This study used the most recent DHS data and a cross-sectional study design. The DHS are national surveys carried out every five years in low-and middle-income countries globally. The surveys concentrate on maternal and child health, physical activity, sexually transmitted infections, fertility, health insurance, tobacco use, and alcohol consumption. They provide data to monitor the demographic and health profiles of the respective countries. For the study, women with birth history who had given birth up to five years before the survey were included. Only the last birth of the women aged 15–49 years preceding the survey was included in the study. A sample of 4,290 women with complete data required for our analysis participated in this study. Permission to use the data set was given by the MEASURE DHS following the assessment of our concept note. This study is a secondary analysis of the de-identified 2014 Ghana Demographic and Health Survey (GDHS), a publicly available dataset. Therefore, ethics approval or consent to participate is not applicable. The datasets are freely available to the public at www.measuredhs.com.

### Description of variables

**Outcome variables.** *Outcome variable*. The primary outcome variable was unskilled birth attendance. The outcome variable was derived from the response to the question "who assisted

with the delivery?" Responses were categorized under Health Personnel and Other Person. Health personnel included doctor, nurse, nurse/midwife, auxiliary midwife, and other people consisting of a traditional birth attendant, traditional health volunteer, community/village health volunteer, neighbours/ friends/relatives, etc. For this study, unskilled birth attendance referred to births assisted by a traditional birth attendant, traditional health volunteer, community/village health volunteer, neighbours/ friends/relatives, other [11].

*Explanatory variables.* Eleven explanatory variables were used. These were grouped into individual and community level variables. The individual characteristics consist of level of education, age, parity, health insurance coverage, wealth status, distance to a health facility, and media exposure. The community-level characteristics comprised the type of residence, community socioeconomic status, community literacy, and region of residence.

## Statistical analysis

We employed both descriptive and inferential analytical approaches. First, we computed the proportion of women who utilized the service of unskilled birth attendants during delivery. This ensued with bivariate analysis between individual characteristics (level of education, age, parity, health insurance coverage, wealth status, distance to a health facility, and media exposure), community characteristics (Type of residence, Community socioeconomic status, Community literacy, Region of residence) and utilization of unskilled birth attendants (see Table 1). Following the hierarchical nature of the data set, the Multilevel Logistic Regression Model (MLRM) was employed. This comprises fixed effects and random effects [23]. The fixed effects of the model were gauged with binary logistic regression, which resulted in odds ratios (ORs) and adjusted odds ratios (aORs) (see Table 2). Model 1 was an empty table, where model 2 looked at the relationship between the individual variable and the outcome variable. Model 3 looked at the relationship between community variables and the outcome variable. Model 4 was the complete model that looked at the relationship with both the individual and community variables and the outcome variable. The random effects, on the other hand, were assessed with Intra-Cluster Correlation (ICC) [23] (Table 3). The sample weight (v005/ 1,000,000) was applied in all the analyses to control for over and under-sampling. All the analyses were carried out using STATA version 14.

## Model fit and specifications

We assessed the fitness of all the models with the Likelihood Ratio (LR) test. The presence of multicollinearity between the independent variables was checked before fitting the models. The variance inflation factor (VIF) test revealed the absence of high multicollinearity between the variables (Mean VIF = 2.28).

## Spatial analysis

In the conduct of the survey, instead of mapping outhouses in which the data were collected, clusters were mapped to protect the actual identity and location of respondents [24]. These clusters are developed to suit the district-level data, making it easy to merge the household records with spatial data. During the data collection period, there were 216 administrative districts in Ghana; however, not all districts had respondents drawn from for the survey. This aided in the merger of the data gathered with the district shapefiles obtained from the Department of Geography and Regional Planning, University of Cape Coast, Ghana. This was done to permit the analysis to be made at a district level. The data is best analysed at the district level since the information is more representative at the cluster level [24]. This study extracted the required variables from the 2014 GDHS. The extracted data maintained the mapped clusters

**Table 1. Measurement of variables.**

| No | Variable | Description/Question | Coding |
|----|----------|----------------------|--------|
| | | **Outcome variable** | |
| 1 | Unskilled Birth Attendance | Who assisted with the delivery? | 0 = skilled birth attendant |
| | | | 1 = unskilled birth attendant |
| | | **Explanatory/independent variables** | |
| **Individual variable** | | | |
| 1 | Level of Education | Level of Education | 0 = no education |
| | | | 1 = primary |
| | | | 2 = secondary |
| | | | 3 = higher |
| 2 | Age | Age of respondents | 1 = 15–19 |
| | | | 2 = 20–24 |
| | | | 3 = 25–29 |
| | | | 4 = 30–34 |
| | | | 5 = 35–39 |
| | | | 6 = 40–44 |
| | | | 7 = 45–49 |
| 3 | Parity | Birth history of women | 1 = Primiparous |
| | | | 2 = Multiparous |
| | | | 3 = Grand multiparous |
| 4 | Health insurance coverage | Covered by health insurance | 0 = no |
| | | | 1 = yes |
| 5 | Getting medical help for self: distance to a health facility | Whether the respondents have a problem with distance, getting medical help | 1 = big problem |
| | | | 2 = not a problem |
| 6 | Media exposure | Media exposure looks at the exposure of respondents to listening to the radio, watching television, and reading the newspaper/magazine | 1 = low |
| | | | 2 = medium |
| | | | 3 = high |
| Community variables | | | |
| 7 | Type of residence | Type of residence | 1 = urban |
| | | | 2 = rural |
| 8 | Community socioeconomic status | Community socioeconomic status | 1 = low |
| | | | 2 = high |
| 9 | Community literacy | Community literacy | 1 = low |
| | | | 2 = medium |
| | | | 3 = high |
| 10 | Region of residence | Region of residence of respondents | 1 = Western |
| | | | 2 = Central |
| | | | 3 = Greater Accra |
| | | | 4 = Volta |
| | | | 5 = Eastern |
| | | | 6 = Ashanti |
| | | | 7 = Brong Ahafo |
| | | | 8 = Northern |
| | | | 9 = Upper East |
| | | | 10 = Upper West |

Source: GDHS, 2014

**Table 2. Background characteristics and unskilled birth attendant.**

| Variable | Frequency N = 4,290 | Percentage (%) | Proportion of unskilled birth attendant |
|---|---|---|---|
| **Individual Characteristics** | | | |
| *Level of education* | | | |
| No education | 1,117 | 26.0 | 44.7 |
| Primary | 822 | 19.7 | 29.9 |
| Secondary | 2,133 | 49.7 | 13.0 |
| Higher | 198 | 4.6 | 1.3 |
| *Age* | | | |
| 15–19 | 190 | 4.5 | 21.1 |
| 20–24 | 728 | 17.0 | 25.5 |
| 25–29 | 1,037 | 24.2 | 22.0 |
| 30–34 | 1,006 | 23.4 | 25.1 |
| 35–39 | 808 | 18.8 | 20.5 |
| 40–44 | 396 | 9.2 | 27.8 |
| 45–49 | 125 | 2.9 | 39.3 |
| *Parity* | | | |
| Primiparous | 977 | 22.8 | 13.2 |
| Multiparous | 2,231 | 52.0 | 22.4 |
| Grand multiparous | 1,982 | 25.2 | 37.3 |
| *Covered by health insurance* | | | |
| No | 1,425 | 33.2 | 31.3 |
| Yes | 2,865 | 66.8 | 20.4 |
| *Wealth status* | | | |
| Poorest | 901 | 21.0 | 50.4 |
| Poorer | 871 | 20.3 | 37.3 |
| Middle | 857 | 20.0 | 21.7 |
| Richer | 844 | 19.7 | 5.1 |
| Richest | 817 | 19.0 | 2.8 |
| *Getting medical help for self: distance to a health facility* | | | |
| Big problem | 1,138 | 26.5 | 37.4 |
| Not a big problem | 3,152 | 73.5 | 19.2 |
| *Media exposure* | | | |
| Low | 405 | | 49.5 |
| Medium | 1,107 | | 33.2 |
| High | 2,778 | | 16.7 |
| **Community variables** | | | |
| *Type of residence* | | | |
| Urban | 1,981 | 46.2 | 8.7 |
| Rural | 2,309 | 53.8 | 37.2 |
| *Community socioeconomic status* | | | |
| Low | 2,953 | 68.8 | 35.6 |
| High | 1,337 | 31.2 | 7.3 |
| *Community literacy* | | | |
| Low | 1,440 | 33.6 | 44.8 |
| Medium | 1,437 | 33.5 | 26.3 |
| High | 1,413 | 32.9 | 8.3 |
| *Region of residence* | | | |
| Western | 442 | 10.3 | 22.3 |

(*Continued*)

**Table 2.** (Continued)

| Variable | Frequency N = 4,290 | Percentage (%) | Proportion of unskilled birth attendant |
|---|---|---|---|
| Central | 472 | 11.0 | 27.3 |
| Greater Accra | 698 | 16.3 | 6.8 |
| Volta | 327 | 7.6 | 29.1 |
| Eastern | 403 | 9.4 | 30.7 |
| Ashanti | 764 | 17.8 | 12.4 |
| Brong Ahafo | 388 | 9.0 | 19.7 |
| Northern | 497 | 11.6 | 60.5 |
| Upper east | 184 | 4.3 | 15.2 |
| Upper west | 115 | 2.7 | 32.5 |

Source: GDHS, 2014

information. This mapped cluster information was used to help join the extracted non-spatial data to the coordinates gathered for the clusters. All the data required (GDHS data and 216 district boundary) were projected into the projected coordinate system of Ghana Meter Grid to aid in the spatial analysis. The extracted GDHS data were merged with coordinate, and a spatial join was undertaken to transfer the cluster point to the 216-district boundary (polygon) layer using ArcMap version 10.5. This activity enabled us to easily identify and trace where each case is located within a district. It was identified that some of the district boundaries had more than one cluster. In such cases, the data from the clusters were aggregated, and their means were computed to represent the respective district they fell within [24].

With regards to the geospatial analyses, four spatial statistical tools were applied to analyse the data. These tools were spatial autocorrelation (Global Moran's I), hot spot analysis (Getis-Ord G), outlier and cluster analysis, and Geographically Weighted Regression. The spatial autocorrelation was used to assess whether unskilled birth attendance in Ghana had a clustering or dispersion pattern at the district level. This study hypothesized that unskilled birth attendance is randomly distributed across various districts in the country. The null hypothesis is rejected if a calculated p-value is small (95% confidence interval), which implies an unlikely situation that the observed spatial pattern results from random processes [24]. Further, hot spot analysis (Getis-Ord G) was used to ascertain statistically significant spatial variations in unskilled birth attendance [24, 25]. This analysis was conducted to determine districts with high prevalence against areas of the low prevalence of unskilled birth attendance. In addition, an outlier and cluster analysis was conducted to identify districts that appeared as outliers. Outlier districts could either be a hot spot district that is surrounded by cold spot districts and vice-versa. The geographically weighted regression (GWR) modelling was conducted after ascertaining the hot spot and cluster and outlier analysis of unskilled birth attendance, the geographically weighted regression (GWR) modelling was conducted. This spatial regression modelling was performed to identify which explanatory variables best account for the observed spatial patterns of unskilled birth attendance [25]. To be specific, the GWR uses the OLS coefficient from the clusters concerning its nearest neighbours in modelling the predictability of the explanatory variable. The output shows how the strength of each explanatory variable changed across space. Therefore, maps of the statistically significant coefficients were generated.

**Table 3. Multilevel logistic regression of unskilled birth attendants among women in Ghana.**

| Variables | Model 1 | Model II aOR (95% CI) | Model III aOR (95% CI) | Model IV aOR (95% CI) |
|---|---|---|---|---|
| *Level of education* | | | | |
| No education | | Ref | | Ref |
| Primary | | 0.77**(0.60, 0.97) | | 0.81(0.63, 1.03) |
| Secondary | | 0.49***(0.38, 0.63) | | 0.52***(0.40, 0.67) |
| Higher | | 0.07*(0.01, 0.53) | | 0.07*(0.01, 0.55) |
| *Age* | | | | |
| 15–19 | | 1.01(0.62, 1.66) | | 1.01(0.62, 1.66) |
| 20–24 | | 0.89(0.67, 1.19) | | 0.90(0.67, 1.20) |
| 25–29 | | Ref | | Ref |
| 30–34 | | 0.82(0.63, 1.08) | | 0.87(0.66, 1.14) |
| 35–39 | | 0.60**(0.44, 0.82) | | 0.64**(0.47, 0.87) |
| 40–44 | | 0.59**(0.40, 0.85) | | 0.65**(0.45, 0.94) |
| 45–49 | | 0.71(0.43, 1.16) | | 0.75(0.46, 1.23) |
| *Parity* | | | | |
| Primiparous | | 0.49***(0.36, 0.66) | | 0.51***(0.38, 0.70) |
| Multiparous | | Ref | | Ref |
| Grand multiparous | | 1.30*(1.01, 1.67) | | 1.26(0.97, 1.62) |
| *Covered by health insurance* | | | | |
| No | | 1.81***(1.49, 2.19) | | 1.80***(1.48, 2.19) |
| Yes | | Ref | | Ref |
| *Wealth status* | | | | |
| Poorest | | Ref | | Ref |
| Poorer | | 0.80(0.63, 1.03) | | 0.78(0.60, 1.01) |
| Middle | | 0.46***(0.34, 0.63) | | 0.53***(0.38, 0.70) |
| Richer | | 0.10***(0.06, 0.16) | | 0.16***(0.10, 0.27) |
| Richest | | 0.08***(0.04, 0.16) | | 0.17***(0.08, 0.36) |
| *Getting medical help for self: distance to health facility* | | | | |
| Big problem | | 1.21(0.99, 1.48) | | 1.10(0.90, 1.35) |
| Not a big problem | | Ref | | Ref |
| *Media exposure* | | | | |
| Low | | 1.42*(1.06, 1.89) | | 1.35*(1.01, 1.80) |
| Moderate | | 0.99(0.80, 1.24) | | 0.96(0.77, 1.20) |
| High | | Ref | | Ref |
| Community level | | | | |
| *Place of Residence* | | | | |
| Urban | | | 0.33***(0.23, 0.48) | 0.52**(0.35, 0.77) |
| Rural | | | Ref | Ref |
| *Community socioeconomic status* | | | | |
| Low | | | Ref | Ref |
| High | | | 0.48**(0.30, 0.77) | 0.83(0.50, 1.34) |
| *Community literacy* | | | | |
| Low | | | 3.91***(2.51, 6.10) | 2.06**(1.32, 3.23) |
| Medium | | | 2.25***(1.54, 3.28) | 1.60*(1.10, 2.34) |
| High | | | Ref | Ref |
| *Region of residence* | | | | |
| Western | | | 0.99(0.49, 2.04) | 0.92(0.45, 1.89) |
| Central | | | 1.44(0.72, 2.89) | 1.07(0.53, 2.17) |

(*Continued*)

**Table 3.** (Continued)

| Variables | Model 1 | Model II aOR (95% CI) | Model III aOR (95% CI) | Model IV aOR (95% CI) |
|---|---|---|---|---|
| Greater Accra | | | Ref | Ref |
| Volta | | | 1.75(0.85, 3.60) | 1.17(0.56, 2.37) |
| Eastern | | | 1.64(0.81, 3.32) | 1.16(0.56, 2.37) |
| Ashanti | | | 0.72(0.35, 1.48) | 0.59(0.28, 1.23) |
| Brong Ahafo | | | 0.62(0.30, 1.29) | 0.42*(0.21, 0.91) |
| Northern | | | 3.13**(1.15, 6.47) | 1.58(0.75, 3.36) |
| Upper east | | | 0.40*(0.19, 0.85) | 0.20***(0.09, 0.43) |
| Upper west | | | 0.79(0.37, 1.69) | 0.49(0.22, 1.07) |
| **Random effect result** | | | | |
| PSU variance (95% CI) | 2.65(2.09, 3.35) | 1.17(0.89, 1.54) | 0.85(0.62, 1.15) | 0.71(0.51, 0.98) |
| ICC | 0.45 | 0.26 | 0.20 | 0.18 |
| LR Test | $\chi^2 = 854.06$ | $\chi^2 = 281.75$ | $\chi^2 = 185.05$ | $\chi^2 = 130.24$ |
| | p = 0.0000 | p = 0.0000 | p = 0.0000 | p = 0.0000 |
| Wald chi-square | | 356.45 | 302.07 | 463.11 |
| Model fitness | | | | |
| Log-likelihood | -2082.78 | -1864.82 | -1940.67 | -1810.84 |
| BIC | 4182.28 | 3905.29 | 4006.80 | 3906.06 |
| AIC | 4169.55 | 3771.65 | 3911.33 | 3690 |
| N | 4,290 | 4,290 | 4,290 | 4,290 |

*p<0.05

**p<0.01

*** p<0.001

Source: GDHS, 2014

## Results

### Background characteristics and unskilled birth attendant

Forty–four per cent of women with no education utilized the service of an unskilled birth attendant during delivery. A high proportion of women with grand multiparous (37.3%), poorest wealth status (50.4%), low media exposure (49.5%), and those who saw the distance to a health facility as a big problem (37.4%) utilized the service of an unskilled birth attendant during delivery (Table 2). Three in ten women from rural areas and communities with low socioeconomic status utilized the service of an unskilled attendant during delivery (Table 2).

### Multilevel logistic regression of unskilled birth attendants

Results for all the models were presented in Table 2. From the final model (Model IV), our study found out that level of education, age, parity, health insurance coverage, wealth status, media exposure, place of residence, community literacy had a significant relationship with unskilled birth attendants.

Women with higher education had a lesser likelihood (aOR = 0.07, CI = 0.01, 0.55) of utilizing the services of unskilled birth attendants compared to those with no education. Primiparous women have a lesser odd (aOR = 0.51, CI = 0.38, 0.70) utilization of the service of unskilled birth attendants during delivery compared to multiparous women (see Table 2). Women aged 35–39 years were less likely (aOR = 0.64, CI = 0.47, 0.87) to utilize the service of unskilled birth attendants compared to those 25–29 years. Women who were not covered by

health insurance had a higher likelihood of utilizing delivery services from unskilled birth attendants (aOR = 1.80, CI = 1.48, 2.19) compared to those covered by health insurance. We also found that women with the richest wealth status had a lesser odd of utilizing delivery services from unskilled birth attendants (aOR = 0.17, CI = 0.08, 0.36) compared to those with the most deficient wealth status (Table 3). Women with low media exposure had a higher likelihood of utilizing delivery services from unskilled birth attendants (aOR = 1.35, CI = 1.01, 1.80) compared to those with high media exposure (Table 3).

Furthermore, women from urban centres had a lesser odd of utilizing delivery services from unskilled birth attendants (aOR = 0.52, CI = 0.35, 0.77) compared to those from the rural areas. Women from communities with low literacy had a higher chance of utilizing delivery services from unskilled birth attendants (aOR = 2.06, CI = 1.32, 3.23) compared to those from communities with high literacy levels (Table 3).

## Spatial distribution on unskilled birth attendance

The spatial distribution of unskilled birth attendance results from Moran's I spatial autocorrelation analysis (S1 Appendix) revealed that unattended skilled birth was clustered. This implies that unskilled birth attendance was not randomized. Therefore, the non-randomised distribution of unskilled birth attendance led to exploring further to visualize the distribution of unskilled birth attendance in Ghana using Hotspot Analysis (GetisOrd Gi).

The hotspot analysis shows areas of statistically significant high and low intensity of the distribution of a phenomenon under study. According to Fig 1, there is a statistically significant (99% confidence level) high clustering of unskilled birth attendance in some parts of northeastern Ghana (areas shown in red colour). This comprises districts such as Mion, Savelugu-Nanton, Nanimba North, Gushiegu, Kpandai, Zabzugu, Nanumba South, Saboba, Nkwanta South, Yendi Municipal, Karaga, Nkwanta North and Krachi Bchumuru. However, some parts of the south comprising of districts such as Akwapim North, New Juaben Municipal, Shai Osudoku, Ga South Municipal, La-Nkwantanang-Madina, Adenta Municipal, Kpone Katamanso, Accra Metropolis, Lower Manya Krobo, Ashaiman Municipal, Tema Metropolis, Ningo Prampram, Ga East Municipal and La-Dade-Kotopon Municipal showed a cold spot indicating high skilled birth attendance (areas shown in blue colour). Although the hotspot analysis identified statistically significant clustered areas, the cluster and outlier analysis was conducted to show districts with high unskilled birth attendance but were not captured in the hotspot analysis. The hotspot analysis does a cluster analysis and zones areas as hot or cold spots when an area has a similar occurrence of a phenomenon with its neighbours. This necessitated the conduct of the cluster and outlier analysis to identify outlier areas.

The cluster and outlier analysis (Fig 2) revealed that about 12 districts (marked in the colour red); Ahafo Ano South, Ayensuano, Shai Osudoku, Wa East, Jaman North, Tano North, Atwima Nwabiagya, Wassa Amenfi East, Bosome Freho, Sekyere South, Lower Manya Krobo and Ho West, that are randomly distributed have high unskilled birth attendance but could not be captured as a hotspot since they were found closer to areas with low unskilled birth attendance. For instance, Shai Osudoku and Lower Manya Krobo, identified as cold spot after the cluster and outlier analysis, have revealed that they have high unskilled birth attendance but are surrounded by neighbours with high skilled birth attendance. Also, two districts (marked in colour blue), Krachi Nchumuru and Sene West, were found to have low unskilled birth attendance but closer to districts with high unskilled birth attendance. This result reveals that relying on hotspot analysis is reasonable when considering areas with statistically significant similar characteristics to their neighbours, whereas cluster and outlier analysis aids in identifying areas statistically significant with distinctive characteristics.

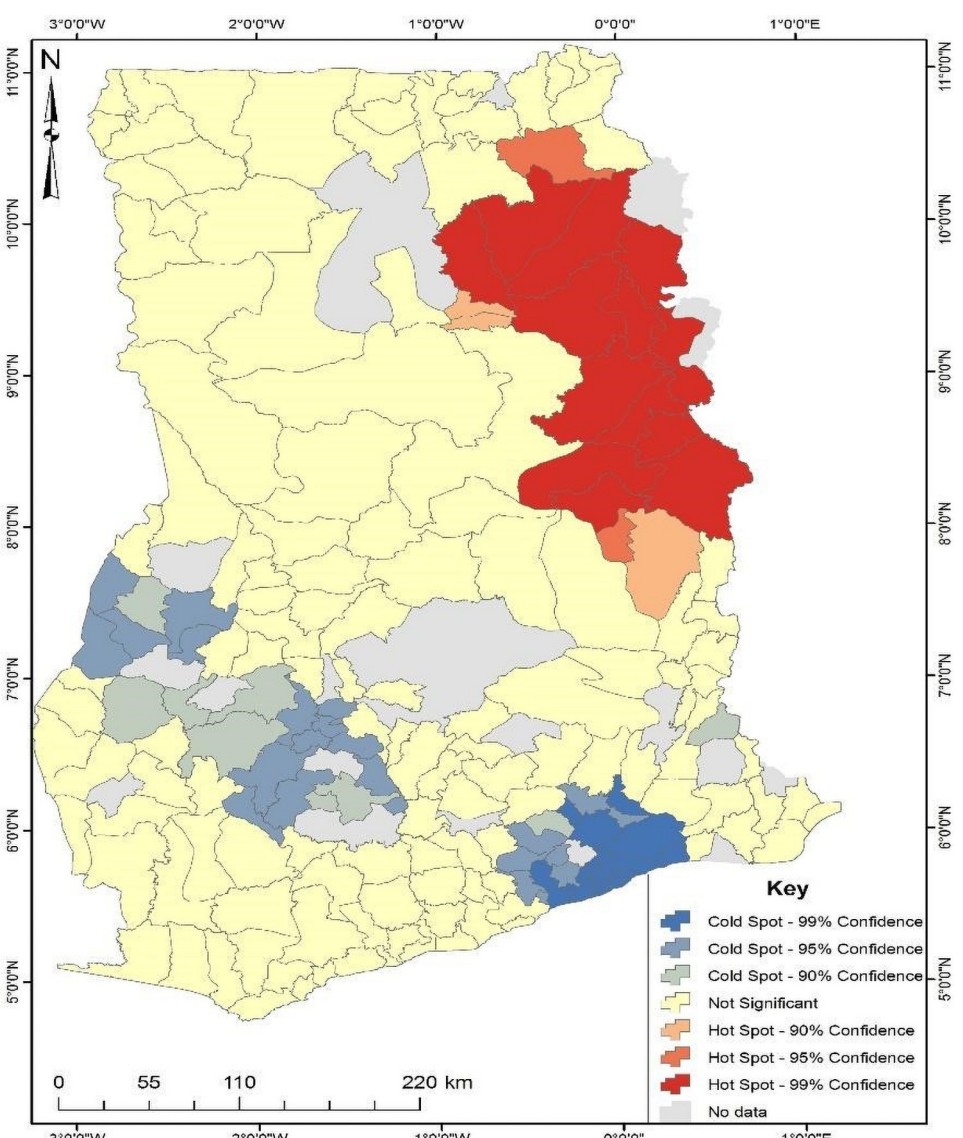

**Fig 1. Hotspot analysis of unskilled birth attendance in Ghana.** Source: Authors' construct (2021).

Further analysis was conducted to assess factors that account for the spatial distribution of unskilled attendance in Ghana. These factors were assessed using the Geographically Weighted Regression (GWR) analysis. However, to conduct GWR, the ordinary least square (OLS) analysis was conducted to identify significant factors that explain the existing distribution of unskilled birth attendance in Ghana. The robust probability results of the OLS (Table 4) revealed that four out of five factors could geographically explain the spatial distribution of unskilled birth attendance in Ghana.

The four significant predictors identified from the OLS results were used to conduct the Geographically Weighted Regression (GWR) analysis (S2 Appendix). The GWR is a local model that estimates the different responses of the dependent variable on independent variables based on locational effects. Table 3 shows the GWR model for unskilled birth attendance in Ghana. Since GWR localizes the results of a phenomenon under study, it improves the

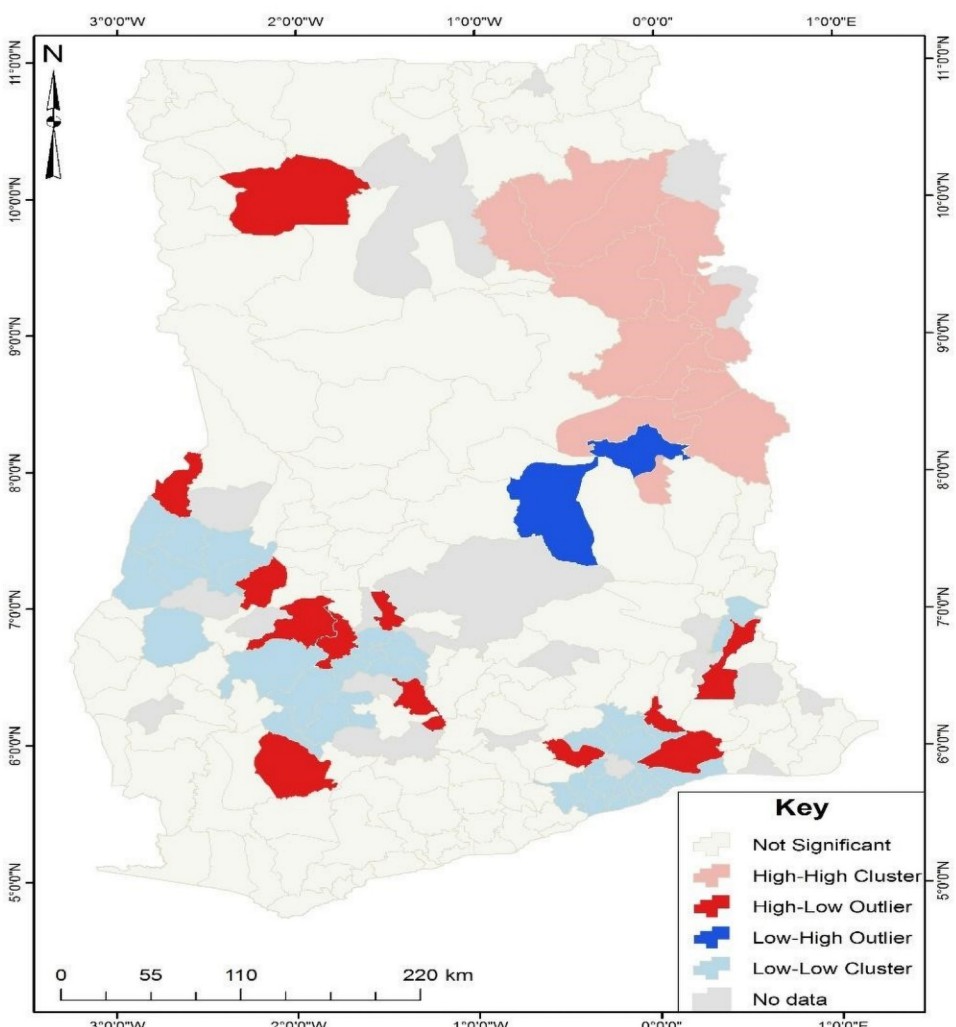

**Fig 2. Cluster and outlier analysis of unskilled birth attendance in Ghana.** Source: Authors' construct (2021).

model fit when the relationship between the predictors and short birth interval is non-stationary [25]. This is evident from the $R^2$ and adjusted $R^2$ values where the OLS value were 0.352 and 0.335 while the GWR values were 0.623 and 0.509, respectively. The GWR's adjust $R^2$ value explains about 50% of the phenomenon under study (S3 Appendix). The subsequent

**Table 4. OLS of factors that accounts for unskilled birth attendance in Ghana.**

| Variable | Coefficient | StdError | t-Statistic | Probability | Robust_SE | Robust_T | Robust Probability |
|---|---|---|---|---|---|---|---|
| Intercept | -0.013 | 0.038 | -0.335 | 0.738 | 0.031 | -0.409 | 0.683 |
| Distance to Facility | 0.272 | 0.057 | 4.789 | 0.000* | 0.062 | 4.403 | 0.000* |
| Media exposure | 0.214 | 0.101 | 2.118 | 0.036* | 0.121 | 1.767 | 0.079 |
| NHIS Subscription | 0.193 | 0.068 | 2.837 | 0.0053* | 0.068 | 2.86 | 0.005* |
| Community economic status | 0.122 | 0.043 | 2.857 | 0.005* | 0.032 | 3.761 | 0.000* |
| Community literacy | 0.097 | 0.037 | 2.619 | 0.010* | 0.046 | 2.115 | 0.036* |

Source: GDHS, 2014.

Figs 3–6 show the spatial predictive power of the explanatory variables towards the unskilled birth attendance in Ghana. Areas shaded red have a strong predictive power, while areas shaded blue have a low predictive power of the explanatory variable.

Regarding the distance to a health facility (Fig 3), the north-eastern part of Ghana (shaded red) had a high positive relationship with unskilled birth attendance. This implies that as the proportion of women who identified distance to a health facility as a problem increased, the occurrence of unskilled birth attendance in the north-eastern part of Ghana increases about 44%-77%.

Also, media exposure was a strong predictor of unskilled birth from the north-east through to the mild belt of Ghana (Fig 4). A unit decrease in media exposure would result in about a 30 to 65% increase in unskilled birth from the north-east through to the mild belt of Ghana.

National health insurance subscription was a predictor from west of the northern region through to the southeast of Ghana (Fig 5). A unit increase in national health insurance subscriptions will result in about a 49 to 72% increase in skilled birth attendance in Ghana.

Finally, it was found that community literacy level serves as a high predictor of unskilled birth attendance in the north-eastern and southwestern parts of Ghana (Fig 6). As community literacy reduces, unskilled birth attendance increases by about 17 to 25%.

## Discussion

Among the variables which adjusted for in the ordinary least squares (OLS), the study found that distance to health facility, national health insurance subscription, community economic status, and community literacy as strong predictors of unskilled birth attendance in Ghana for the spatial analysis (i.e. OLSM) while maternal educational level and age, parity, national health insurance subscription, household wealth, media exposure, place of residence, community-level literacy and region of residence were key findings associated with unskilled birth attendance in the non-spatial model. The nonspatial model explained the variation and clustering that exist in unskilled birth attendance in Ghana. For this reason, appreciating the spatial pattern of the analysis of unskilled birth attendance was deemed essential since it helps visualise the phenomenon under study. Therefore, to complement our analysis, we fitted a spatial model to identify the established pattern further and to enable us to provide some justification for the variation in unskilled birth attendance across the country. Generally, the findings (the predictor variables found in the GWR analysis) were like the multilevel analysis conducted in this study. However, our discussions will be based on the results of the spatial model (GWRM) because that is the focus of this study.

The study's findings suggested that unskilled birth attendance has a significant spatial variation in Ghana. The hot spot analysis found a total of 16 statistically significant districts at various levels of significance with a high prevalence of unskilled birth attendance. Significant hotspots of unskilled birth attendance were seen in the north-eastern part of Ghana. This finding is in line with findings from Dickson and Amu [20]. They reported that women in the Northern and Upper West regions of Ghana were less likely to access skilled assistance during delivery. Variations in skilled birth attendance have been reported in most developing countries [26, 27]. These variations could be due to the physical inaccessibility and several socioeconomic hardships [16], coupled with inequalities in the distribution of scarce resources [17], including inadequately skilled health professionals in remote/border areas of Ghana. In addition, socio-cultural beliefs and practices among women in different districts are potential drivers for the prevalence of unskilled birth participation in Ghana [27, 28]. Furthermore, mothers from border districts may have limited access to information concerning maternal health services and other services such as access to school/education that could affect their chances of

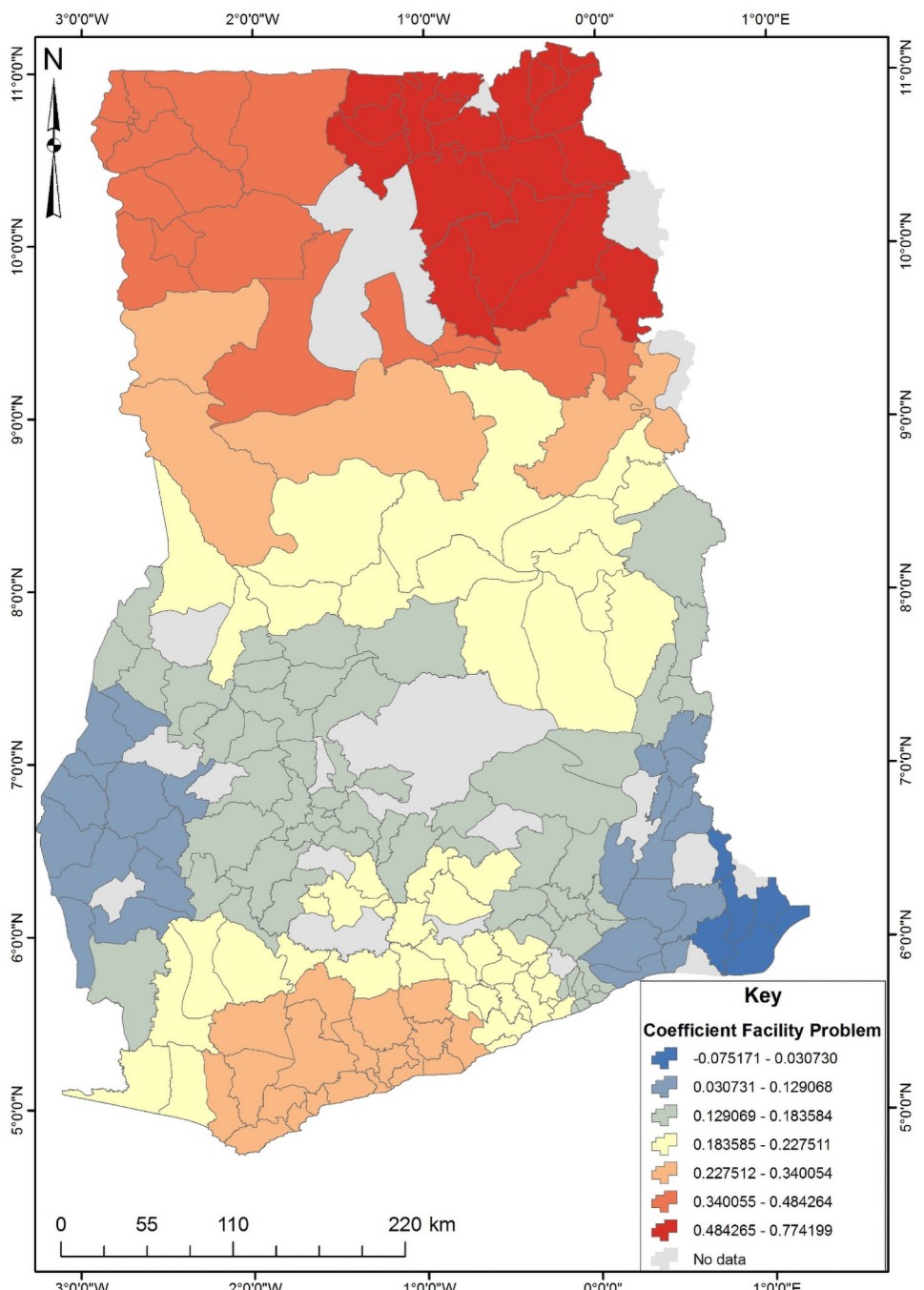

**Fig 3. Distance to facility GWR coefficient for predicting unskilled birth attendance in Ghana.** Source: Authors' construct (2021).

accessing the services of the skilled birth attendant, as was also reported by Antwi and his colleagues [9].

The cluster and outlier analysis was used to verify and complement the hot spot analysis as it allows us to detect both groupings and areas where anomalies exist [29]. The cluster and outlier analysis results show aspects that may have been overlooked in the hot spot analysis but

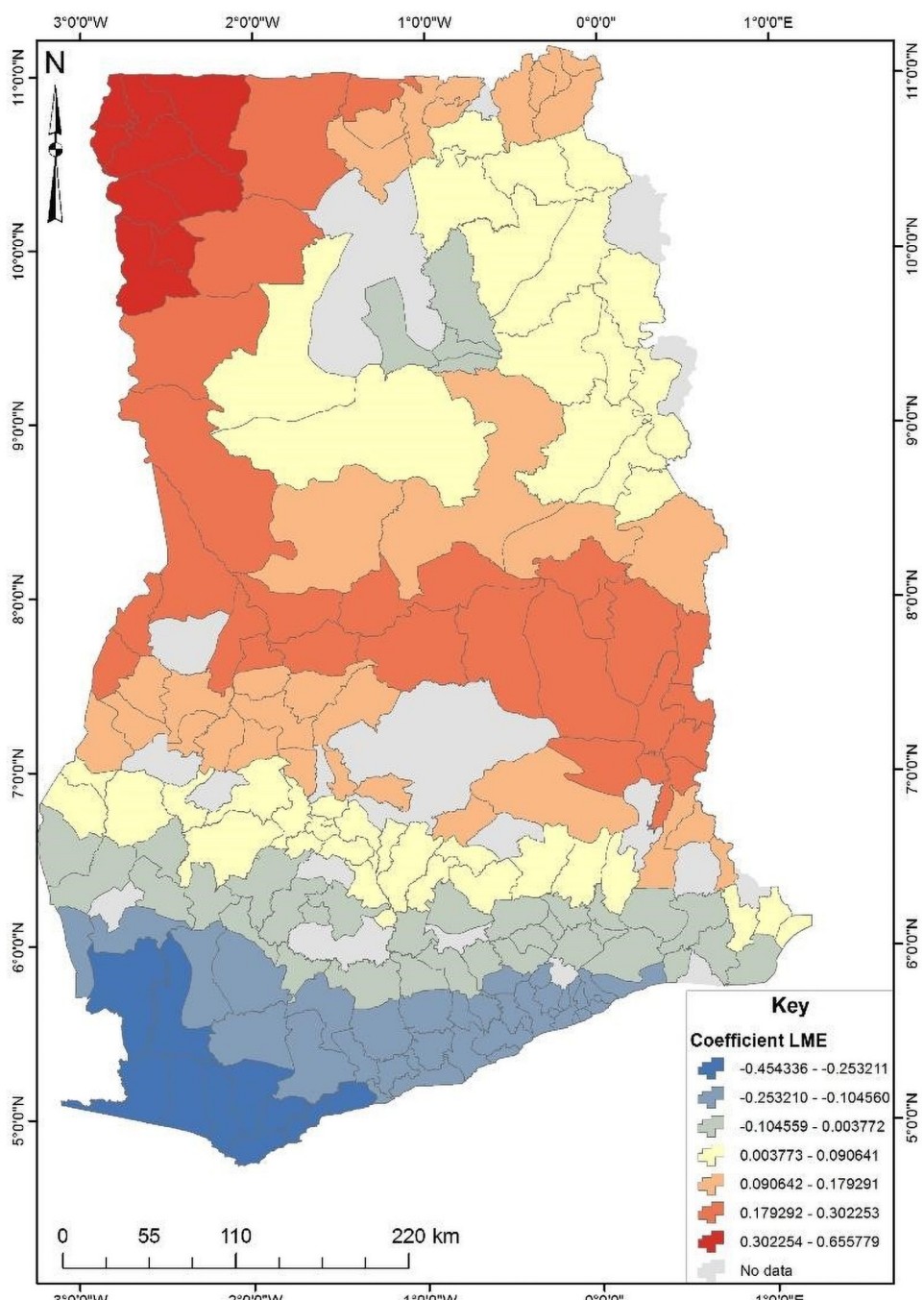

**Fig 4. Media exposure GWR coefficient for predicting unskilled birth attendance in Ghana.** Source: Authors'
construct (2021).

are interesting highlights, especially in those areas where diverse types of groupings coexist to
influence the study's outcome. Districts labelled high-low outliers are anomalous districts with
a high hotspot for unskilled birth attendance, but districts with high skilled assisted deliveries
surround them. The reason for this outcome is rooted in the uneven distribution of scarce

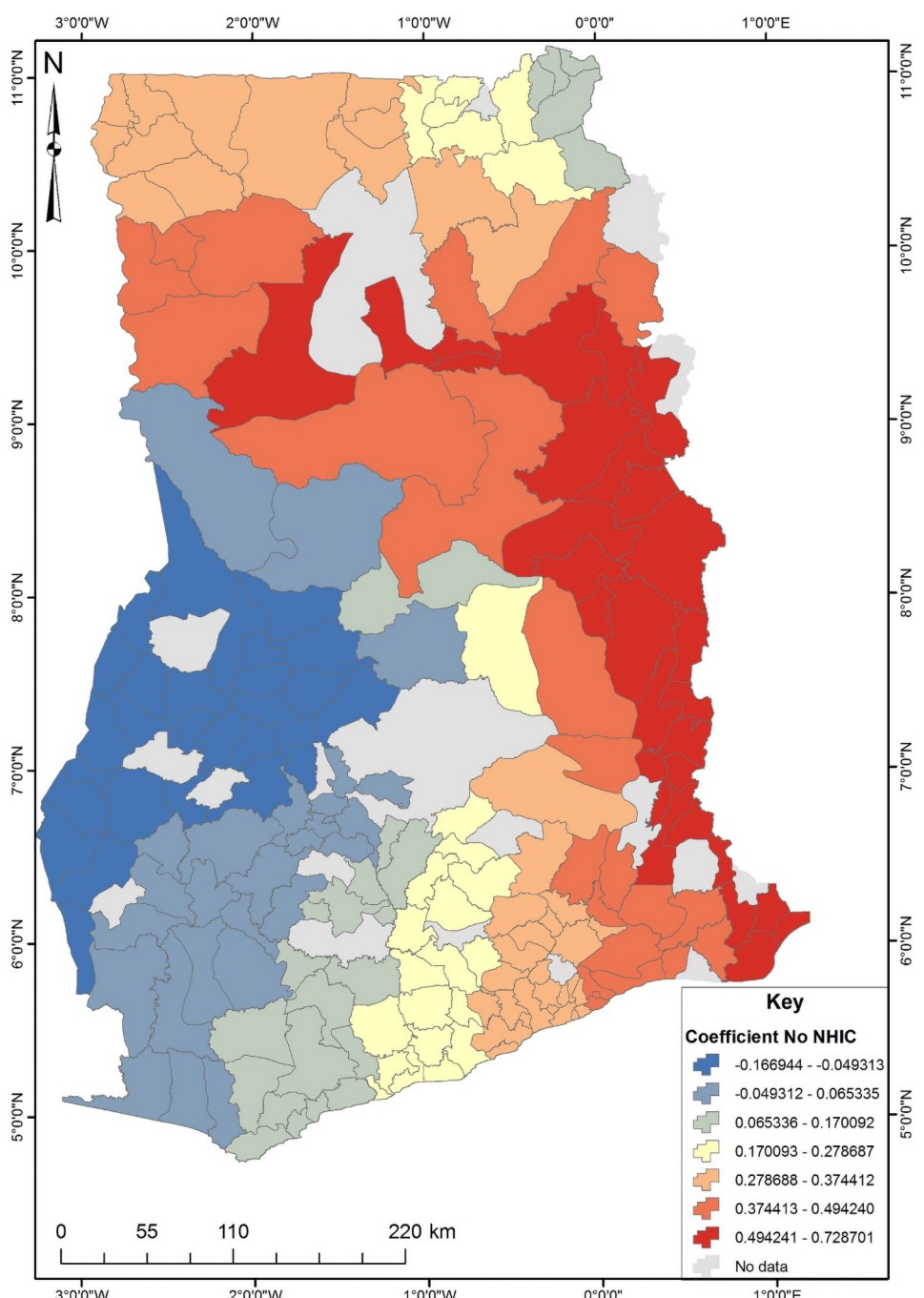

**Fig 5. NHIS subscription GWR coefficient for predicting unskilled birth attendance in Ghana.** Source: Authors' construct (2021).

resources. Most areas with fewer populations do not get the needed infrastructural facilities and personnel since some professionals refuse to post to rural communities in Ghana [12]. A similar explanation can be extended to low-high outlier areas were districts with high unskilled birth attendance surround districts with low unskilled birth attendance.

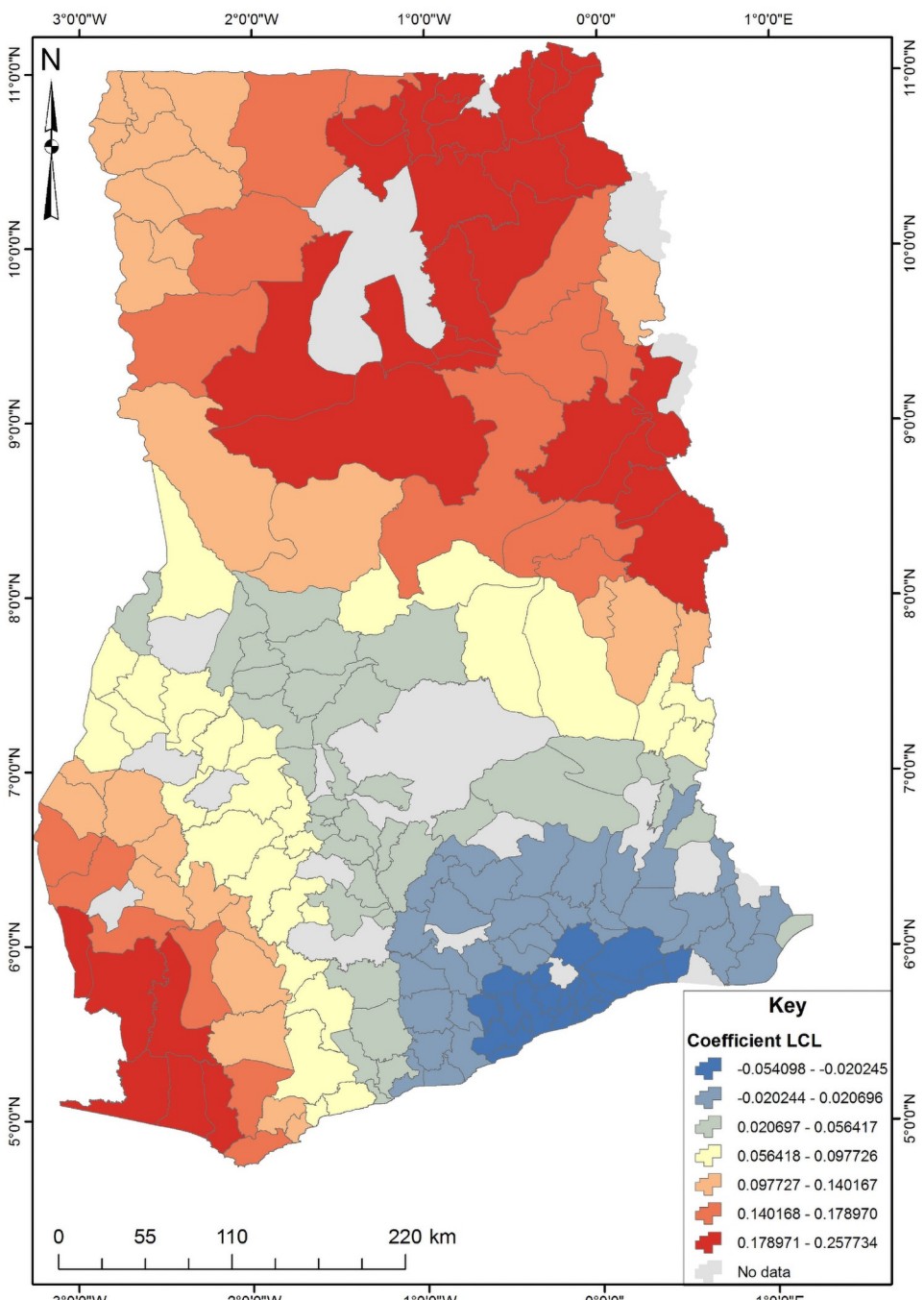

**Fig 6. Community literacy level GWR coefficient for predicting unskilled birth in Ghana.** Source: Authors'
construct (2021).

The GWR analysis revealed a strong positive relationship between mothers who perceived
distance to the health facility as a big problem, those not covered by health insurance, those
who resided in communities with low literacy and socio-economic levels with unskilled birth
attendance in different districts in Ghana. However, exposure to social media could not pre-
dict unskilled birth attendance among various districts in the country. There was a higher

likelihood of not visiting the health care facility for those women who were not covered by health insurance in the hotspot districts. This finding is in line with studies conducted in Ghana [30], and elsewhere including Ethiopia [10] and Kenya [27]. A possible explanation could be because of additional costs incurred if payment for healthcare services is made out of pocket. Similarly, women who are not covered by health insurance in hotspot areas are highly likely to seek alternative health care services different from their peers from cold spot areas. Our study supports this—areas with higher proportions of people with health insurance subscriptions tend to be cold spots for unskilled births.

The study further found that district-level characteristics are key influencers of unskilled birth delivery in Ghana. A corroboratory to NHIS coverage was the finding that women residing in districts with low socio-economic levels are more likely to access the services of unqualified and unskilled birth attendants than women residing in districts with medium or higher socio-economic levels. This finding is supported by a study done in Nigeria, where the utilisation of unskilled birth attendance services was high among women living in rural communities with low socioeconomic status [26] and, Ethiopia where women from the poorer and poorest households used less skilled assistance during delivery [27]. Generally, women in districts with higher poverty levels are less likely to pay for health insurance subscriptions [31]. The unavailability of health care facilities to readily cater to the health needs of women in poorer districts significantly influences their utilisation of SBA services.

Moreover, participants who perceived distance from the health facility in hotspot districts as a major problem were less likely to seek services for skilled attendant delivery. This is in line with studies done in Ethiopia [27] and rural Nigeria [26]. This is because the distance to and from health facilities and lack of transport are crucial factors preventing mothers from seeking and using skilled maternity services.

## Conclusion

In Ghana, unskilled birth attendance had spatial variations and clustering across different parts of the country. About 13 randomly distributed districts have high unskilled birth attendance. Districts with high proportions of unskilled birth attendance were detected in the North-Eastern parts of the country. In the GWR analysis, districts with higher proportions of low socio-economic class, low literacy status, highly perceived distance to and from the health facility, and not covered by health insurance were found to have an increased probability of influencing unskilled birth attendance. Due to such spatial variations and clustering, interventions aimed at promoting skilled birth attendance should include primary health care accessibility, particularly at the north-eastern part of Ghana, increase in media coverage, particularly at the north-eastern part through to the middle belt of Ghana, increase in national health insurance subscription particularly at the northern through to the south eastern part of Ghana, and improvement in community literacy level particularly at the north eastern and southwestern parts of Ghana.

## Supporting information

**S1 Appendix. Spatial autocorrelation of unskilled birth in Ghana.** Source: GDHS, 2014.
(DOCX)

**S2 Appendix. OLS diagnostics.** Source: GDHS, 2014.
(DOCX)

**S3 Appendix. GWR model for unskilled birth attendance in Ghana.** Source: GDHS, 2014.
(DOCX)

## Acknowledgments

We acknowledge Measure DHS for providing us with the data. Our appreciation goes to Ferdinard Fosu-Blankson for proofreading this article.

## Author Contributions

**Conceptualization:** Vincent Bio Bediako, Bernard Afriyie Owusu.

**Data curation:** Vincent Bio Bediako.

**Formal analysis:** Ebenezer N. K. Boateng, Kwamena Sekyi Dickson.

**Visualization:** Ebenezer N. K. Boateng.

**Writing – original draft:** Kwamena Sekyi Dickson.

**Writing – review & editing:** Bernard Afriyie Owusu, Kwamena Sekyi Dickson.

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
