## [Decision Letter · Decision Letter 0]

7 May 2021

PONE-D-21-07780

Multilevel Geospatial Analysis of unskilled birth attendant in Ghana

PLOS ONE

Dear Dr. Bediako,

Thank you for submitting your manuscript to PLOS ONE. After careful consideration, we feel that it has merit but does not fully meet PLOS ONE’s publication criteria as it currently stands. Therefore, we invite you to submit a revised version of the manuscript that addresses the points raised during the review process.

We look forward to receiving your revised manuscript.

Kind regards,

Kannan Navaneetham, PhD

Academic Editor

PLOS ONE

Journal Requirements:

4. We note that Figures 1-7 in your submission contain map images which may be copyrighted. All PLOS content is published under the Creative Commons Attribution License (CC BY 4.0), which means that the manuscript, images, and Supporting Information files will be freely available online, and any third party is permitted to access, download, copy, distribute, and use these materials in any way, even commercially, with proper attribution. For these reasons, we cannot publish previously copyrighted maps or satellite images created using proprietary data, such as Google software (Google Maps, Street View, and Earth). For more information, see our copyright guidelines: http://journals.plos.org/plosone/s/licenses-and-copyright.

4.1.    You may seek permission from the original copyright holder of Figures 1-7 to publish the content specifically under the CC BY 4.0 license. 

4.2.    If you are unable to obtain permission from the original copyright holder to publish these figures under the CC BY 4.0 license or if the copyright holder’s requirements are incompatible with the CC BY 4.0 license, please either i) remove the figure or ii) supply a replacement figure that complies with the CC BY 4.0 license. Please check copyright information on all replacement figures and update the figure caption with source information. If applicable, please specify in the figure caption text when a figure is similar but not identical to the original image and is therefore for illustrative purposes only.

Reviewers' comments:

Reviewer's Responses to Questions

**Comments to the Author**

1. Is the manuscript technically sound, and do the data support the conclusions?

Reviewer #1: Yes

Reviewer #2: Yes

2. Has the statistical analysis been performed appropriately and rigorously? 

Reviewer #1: Yes

Reviewer #2: Yes

3. Have the authors made all data underlying the findings in their manuscript fully available?

Reviewer #1: Yes

Reviewer #2: Yes

4. Is the manuscript presented in an intelligible fashion and written in standard English?

Reviewer #1: No

Reviewer #2: Yes

5. Review Comments to the Author

Reviewer #1: Reviewer’s Comments

Multilevel geospatial analysis of unskilled birth attendant in Ghana

The authors examined an important aspect maternal health especially in Ghana. However, I have some few comments for consideration.

General observations

The topic should be revised to “Multilevel geospatial analysis of factors associated with unskilled birth deliveries in Ghana”

The manuscript requires proofreading to improve upon the English Language

Referencing has been done haphazardly. There is no coherence right from the start of writing. E.g. in the Background, the first reference was numbered (41), then followed with (36) and (13). This inconsistency is obvious throughout the whole work.

Background

Paragraphs 2 and part of paragraph 3 should not be part of the background. These paragraphs highlights issues on the definition of skilled births and some related statistics of skilled births. The focus of the study is unskilled births. Therefore, the background should succinctly concentrate on unskilled births. This will distinguish it from studies that have examined skilled births in Ghana.

Also, the opening sentence of the last paragraph of the background suggests a special interest in a particular ecological zone in Ghana. This I believe is the case. The study considered the country. This too needs to be revised. Rather highlight unskilled birth situations in the country.

The whole background should be revised in a coherent manner to inform the reader exactly what the study is meant to examine, thus unskilled births in Ghana.

Methods

Source of data

“For the study, women with birth history who had given birth up to five years before the survey were included”. These inclusion criteria is not clear. With a period of five years, a woman can comfortably give birth to at least two children assuming there are no any medical complications. And these births can occur at home or at health facility or both at home or both at a health facility. So, which one of these children should the study be on? For uniform selection of women, only the last birth of the women aged 15-49 years preceding the survey should be included in the study. If this done, it will certainly affect the current sample size of 4,290 women; meaning re-analysis is required.

Outcome variable

How was the outcome variable coded?

Explanatory variable

It will be interesting for the authors to indicate the various responses of the variables as captured in the GDHS and then indicate how each of them was recoded, where necessary.

Statistical analysis

Authors should indicate how variables were selected for the regression models. It seems preference was given to some variables at each stage of the modelling.

Spatial analysis

This section is too detailed. It should be summarized in two paragraphs for purposes of sustaining the interest of reader and also making the content simple.

Results

Background characteristics

• To refer to a table in text, state it as (Table 1) and not as (see Table 1).

• On Table 1, proportion for ‘Richest’ category of wealth status is missing.

Spatial distribution on unskilled births attendants

Some of the spatial figures and tables can be on supplementary sheets. Some of these include:

a. Fig. 1: Spatial autocorrelation of unskilled births in Ghana

b. Table 4: OLS Diagnostics

c. Table 5: GWR model for unskilled birth attendance in Ghana

The authors failed to mention the specific districts that are associated with unskilled births in Ghana. It is important to mention these districts to better inform policy makers and readers where unskilled births are high within the country.

The discussion was well presented however the referencing style is not appropriate.

Conclusion too was well presented.

Reviewer #2: Authors have adequate literature review and analysis of the data in this study. This study is novel enough to be published.

Small typographical errors and a few unclear sentences to correct. Standardize use of Fig. and Figures. Abbreviations and acronyms should only be spelled out at first use.

6. PLOS authors have the option to publish the peer review history of their article (what does this mean?). If published, this will include your full peer review and any attached files.

Reviewer #1: No

Reviewer #2: No

---

## [Author Response · Author response to Decision Letter 0]

1 Jun 2021

In submitting our revised manuscript entitled “Multilevel Geospatial Analysis of Factors Associated with Unskilled Birth Attendance in Ghana”, we would like to draw the attention of the journal to a copyright issue which is related to the maps used in the manuscript. We state emphatically that all maps in the manuscript were designed by the authors based on the data used for the study. All the authors are in the know and consent to this fact. 

Thank you. 

Thank you.

Vincent Bio Bediako

(Corresponding Author)

---

## [Editor Report · Decision Letter 1]

9 Jun 2021

Multilevel Geospatial Analysis of Factors Associated with Unskilled Birth Attendance in Ghana

PONE-D-21-07780R1

Dear Dr. Bediako,

We’re pleased to inform you that your manuscript has been judged scientifically suitable for publication and will be formally accepted for publication once it meets all outstanding technical requirements.

Kind regards,

Kannan Navaneetham, PhD

Academic Editor

PLOS ONE
---

## [Editor Report · Acceptance letter]

18 Jun 2021

PONE-D-21-07780R1 

Multilevel Geospatial Analysis of Factors Associated with Unskilled Birth Attendance in Ghana. 

Dear Dr. Bediako:

I'm pleased to inform you that your manuscript has been deemed suitable for publication in PLOS ONE. Congratulations! Your manuscript is now with our production department. 

Kind regards, 

on behalf of

Prof. Kannan Navaneetham 

Academic Editor

PLOS ONE